# Assessing Algorithmic Bias in Language-Based Depression Detection: A Comparison of DNN and LLM Approaches

Obed Junias *
*Computer Science*
*University of Colorado Boulder*
Boulder, CO, USA
obed.junias@colorado.edu

Prajakta Kini *
*Computer Science*
*University of Colorado Boulder*
Boulder, CO, USA
prajakta.kini@colorado.edu

Theodora Chaspari
*Inst. Cognitive Science & Computer Science*
*University of Colorado Boulder*
Boulder, CO, USA
theodora.chaspari@colorado.edu

*Abstract*—This paper investigates algorithmic bias in language-based models for automated depression detection, focusing on socio-demographic disparities related to gender and race/ethnicity. Models trained using deep neural networks (DNN) based embeddings are compared to few-shot learning approaches with large language models (LLMs), evaluating both performance and fairness on clinical interview transcripts from the Distress Analysis Interview Corpus/Wizard-of-Oz (DAIC-WOZ). To mitigate bias, fairness-aware loss functions are applied to DNN-based models, while in-context learning with varied prompt framing and shot counts is explored for LLMs. Results indicate that LLMs outperform DNN-based models in depression classification, particularly for underrepresented groups such as Hispanic participants. LLMs also exhibit reduced gender bias compared to DNN-based embeddings, though racial disparities persist. Among fairness-aware techniques for mitigating bias in DNN-based embeddings, the worst-group loss, which is designed to minimize loss for the worst-performing demographic group, achieves a better balance between performance and fairness. In contrast, the fairness-regularized loss minimizes loss across all groups but performs less effectively. In LLMs, guided prompting with ethical framing helps mitigate gender bias in the 1-shot setting. However, increasing the number of shots does not lead to further reductions in disparities. For race/ethnicity, neither prompting strategy nor increasing $N$ in $N$-shot learning effectively reduces disparities.

*Index Terms*—Depression, language, algorithmic bias, deep neural networks, language embeddings, large language models

## I. INTRODUCTION

Advances in natural language processing (NLP) have opened promising avenues for diagnosis and treatment monitoring of depression through identifying linguistic cues in spoken or written language [1]. Prior work suggests that individuals with depression tend to use more first-person singular pronouns, negative emotion words, and absolutist terms [2]–[4]. Their speech may also exhibit greater abstraction, redundancy [5], and lexical repetition [6]. Early approaches to detecting depression from text relied on lexicon-based features [7], [8] and embeddings derived from Word2Vec [9], [10] combined with traditional models [11]. Since these methods often fail to capture contextual semantics, later work adopted

deep contextual embeddings from models such as Bidirectional Encoder Representations from Transformers (BERT) and RoBERTa [12], [13]. More recently, large language models (LLMs) have been explored under zero- and few-shot settings for depression detection [14]–[18].

While advances in NLP have enhanced depression detection, an increasing number of studies highlight persistent demographic biases in such models. Studies analyzing social media content reveal that linguistic features used for depression detection vary across demographic groups [19]. LLMs also exhibit differential performance in detecting depression from spoken data across dimensions such as age, gender, and language [20], as well as in clinical decision-making tasks [21]. These disparities may reflect underlying differences in depression prevalence and the severity of symptoms. For example, there is systematic evidence regarding higher rates of depression among women compared to men [22], [23], and greater severity among Black and Hispanic individuals compared to non-Hispanic White individuals [24]. Disparities may also stem from gender and race-related differences in language use, including differences in vocabulary, grammar, turn-taking, and interruption [25]–[27].

Despite these concerns, most algorithmic fairness assessments in depression detection focus on audio or multi-modal systems [28], [29], with limited attention to text-only models trained on clinical interviews. The widespread adoption of LLMs further complicates this problem, as it remains unclear whether out-of-the-box LLMs amplify existing socio-demographic disparities in the focal task. Research on bias mitigation in ML models for depression detection has primarily focused on tabular data, such as electronic health records [30] and lifestyle or environmental factors [31]. Relatively few studies have examined bias mitigation techniques applied to text data from social media posts [32] or to multimodal data from clinical interviews [33]. While recent work has explored debiasing strategies, including in-context learning, in general-purpose LLMs [34]–[36], the evaluation of these methods in clinical contexts remains unexplored.

This paper investigates algorithmic bias in language-based

---

* These authors contributed equally to this work.

models for depression detection. It compares trade-offs between depression detection performance and algorithmic bias on conventional models trained using DNN-based embeddings with few-shot learning approaches using large language models (LLMs). The study further explores fairness-aware modeling constraints in mitigating bias in language-based depression detection models trained with DNN-based embeddings, and in-context learning for mitigating algorithmic bias in LLM-based depression detection. Experiments are conducted on the Distress Analysis Interview Corpus/Wizard-of-Oz (DAIC-WOZ) dataset [37], with gender and race/ethnicity analyzed as the primary sources of bias. We structure our investigation around the following research questions:

- **RQ1:** How do DNN-based embeddings compare to LLMs in language-based depression detection performance and algorithmic bias?
- **RQ2:** How do different loss functions, implemented as fairness-aware modeling constraints, affect algorithmic bias and depression detection in DNN-based models?
- **RQ3:** How does prompt framing and number of examples in $N$-shot learning influence algorithmic bias and depression detection performance in LLMs?

This paper makes the following contributions to the existing body of knowledge: (1) We compare DNN-based embeddings and LLMs for language-based depression detection, not only in terms of performance, as explored in prior work [14]–[18], but also in terms of algorithmic fairness; (2) We examine various fairness-aware modeling constraints as bias mitigation strategies for text-based depression detection, a task relatively unexplored, with prior work mostly focusing on tabular or multimodal data [30], [31], [33]; and (3) We investigate in-context learning techniques using different prompt framings to mitigate bias in LLMs, an approach that, to our knowledge, has not yet been studied for depression detection.

## II. RELATED WORK

Research on text-based automated depression detection has focused primarily on social media posts and clinical interviews [38]. Literature reviews by William & Suhartono and Babu & Kanaga concluded that early depression detection via social media is feasible, given identifiable linguistic and behavioral patterns [39], [40]. Liu *et al.* reviewed ML techniques for depression detection, highlighting the use of features such as word frequencies, psycholinguistic and emotional terms, temporal references, and topic modeling, used with supervised and unsupervised algorithms [41]. Husseini *et al.* found that convolutional neural networks (CNNs) with max-pooling achieved better performance and generalization on unseen data when compared to recurrent neural networks (RNNs) [42], while Sadeque *et al.* observed better performance from RNNs over non-sequential models [43]. Pre-trained mental health specific models like MentalBERT and MentalRoBERTa have shown further gains in performance [12].

Recent work has begun to explore the capabilities of LLMs for this task. Zero-shot settings show mixed results: some studies suggest that these struggle with domain-specific subtleties [15], while advanced architecture-based models like GPT-4 perform well [14]. Fine-tuning LLMs on mental health-specific data has emerged as a more reliable strategy, given access to high-quality domain-relevant corpora [14], [15]. Beyond classification, chain of thought (CoT) prompting has been used to guide LLM reasoning using established diagnostic criteria [16]; other approaches integrate LLMs with graph-based models [44], or ensemble methods leveraging multiple LLMs to improve knowledge diversity [17].

Despite the encouraging results in automatic depression detection, studies in this domain have raised concerns about demographic bias in mental health (MH) models [45]. Key language markers of depression, such as the use of first-person pronouns, depend on race [19]. ML models trained on conversational data [28], [29] or social media data [19] have exhibited group disparities, with studies suggesting a significant variation in depression detection performance across different groups defined by race and gender. Other studies reveal similar findings for models such as flanT5 and GPT, especially when models are trained on social media datasets with a gender imbalance such as Reddit [20]. Also, models generally achieved higher accuracy for detecting depression in English, likely due to the rich and comprehensive training data available in this language. LLMs have further been evaluated for their ability to provide medical recommendations in terms of MH assessment with results indicating that Black, unhoused, and LGBTQIA+ individuals are more likely to receive recommendations for urgent care, invasive procedures, or mental health assessments compared to non-sensitive groups [21]. High-income patients were more often recommended advanced diagnostic tests compared to low-income patients [21]. Other work has uncovered no socio-demographic bias in terms of psycholinguistic traits and tone of words used in an LLM designed to simulate physician–patient conversations [46].

Review papers outline different types of algorithmic fairness, including individual (i.e., making similar predictions for similar individuals) and group (i.e., making similar predictions for different groups) fairness [47]. This paper focuses on group fairness, specifically evaluating whether text-based depression detection models exhibit equitable performance across demographic subgroups such as gender and race/ethnicity. Debiasing methods are typically categorized into three main types, namely pre-, in-, and post-processing methods, each with varying levels of success in prior work [48]. Yet, there remains a significant gap in addressing algorithmic bias in ML models applied to mental health (MH). Bailey & Plumbley [29] examined a data re-distribution approach for audio-based depression detection, employing sub-sampling to rebalance datasets across subgroups. Other studies using tabular data containing demographic, medical, and behavioral features found that approaches such as sample reweighting across (group, label) combinations, disparate impact removal (i.e., reassigning feature values to minimize bias), and decision-level threshold adaptation can preserve predictive performance while enhancing fairness. Similarly, Park *et al.* demonstrated

that sample reweighting effectively mitigates bias in tabular models for postpartum depression detection [30]. Despite these findings, evaluations of algorithmic bias in MH detection models that rely on language data remain limited.

Bias mitigation in LLMs has mostly been explored for text generation. Dwivedi *et al.* and Wang *et al.* used fairness-aware prompt engineering and in-context learning to reduce bias in LLM-generated content [34], [36]. Fei *et al.* estimated a language model's label bias using random in-domain words from the task corpus, followed by domain-context calibration [35]. To our knowledge, LLM bias mitigation has not been studied for language-based depression detection.

## III. DATA DESCRIPTION

We used the DAIC-WOZ dataset [37], which contains data collected from clinical interviews designed to support the diagnosis of psychological distress conditions. The severity of depression was measured using the Patient Health Questionnaire-8 (PHQ-8). The dataset contains 189 participants. We excluded 3 participants due to missing gender information, resulting in 186 participants for gender-based analysis. The distribution of depression and no depression cases by gender group was: female (23 depression / 62 no depression) and male (19 depression / 82 no depression). For race/ethnicity-based analysis, we excluded 5 participants with missing race information, resulting in 184 participants. The distribution of depression and no-depression classes was: African American (9/51), Hispanic (3/28), and White/Caucasian (21/78).

## IV. METHODOLOGY

### A. Language-based Depression Detection

DNN-based embeddings included Mental-RoBERTa [12], a domain-adapted variant of RoBERTa pre-trained on mental-health–related corpora. To adapt it to our task, we fine-tuned Mental-RoBERTa on the DAIC-WOZ dataset for binary depression classification. To handle long interview transcripts, we applied a chunking strategy in which question–answer pairs were segmented into overlapping windows of 300 words with an 80-word overlap, ensuring that no question-answer pair was split across chunks. These chunks were fed into the model, and the final depression prediction for a participant was obtained using majority voting across the chunk-level predictions. We used a stratified 5-fold cross-validation for evaluation, where each participant appeared in only one fold, ensuring no data leakage across folds. Final performance metrics were averaged across all folds.

GPT-4o-mini served as the LLM due to its state-of-the-art ability to understand linguistic semantics and pragmatics [49]. We asked the LLM to decide whether depression was present based solely on the input text; this is referred to as unguided prompting (UP) (Table I; see Section IV-C for other types of prompts used for bias mitigation). We constructed $N$-shot prompts, $N \in \{1, 3, 5\}$, in which labeled data from $N$ participants were included in the prompt and predictions were generated for the remaining participants. This procedure

was repeated across 20 trials to average out the randomness introduced by the LLMs.

### B. Bias Mitigation in DNN-based Embeddings

The DAIC-WOZ dataset exhibits a notable imbalance in both gender distribution and the depression rates, which can introduce bias in ML models. To address this, we adopted a gender-balancing approach inspired by [29], [50]. To mitigate data imbalance in the gender distribution, we categorized the dataset into four distinct groups: Female-Depression, Female–No Depression, Male-Depression, and Male-No Depression. Each group was down-sampled to 19 samples (i.e., matching the smallest group size) to ensure equal representation. We repeated this stratified sampling process over 10 random trials and reported metrics averaged across trials. To mitigate data imbalance in the race distribution, we did not apply the same resampling strategy used for gender-based analysis, as it would have yielded training sets too small to support a reliable evaluation, because the Hispanic group had only 3 of 28 participants with depression.

To mitigate algorithmic bias in DNN-based embeddings, we implemented in-processing strategies using fairness-aware loss functions. We applied two fairness-aware training strategies to the Mental-RoBERTa model.

*1) Worst-Group Loss:* We adopted a worst-case (max-loss) formulation that penalizes the model based on the subgroup with the highest cross-entropy loss:

$$\mathcal{L}_{\text{fair}} = \max_{s \in S} \text{CE}(P_{x,s}, P_{y,s}) \qquad (1)$$

Here, $S$ denotes the set of subgroups; $\text{CE}(\cdot)$ is the cross-entropy loss; $P_{x,s}$ is the model's predicted distribution for inputs $x$ in subgroup $s$; and $P_{y,s}$ is the corresponding ground-truth label distribution. This encourages the model to improve its performance on the worst-performing group, thereby promoting fairness.

*2) Fairness-Regularized Loss:* We introduced a fairness-weighted loss that incorporates a penalty term proportional to the maximum disparity in subgroup-specific losses:

$$\mathcal{L}_{\text{total}} = \mathcal{L}_{\text{base}} + \lambda \cdot \max_{(i,j)} |\mathcal{L}_i - \mathcal{L}_j| \qquad (2)$$

where $\lambda$ is a fairness regularization coefficient, and $\mathcal{L}_i$ and $\mathcal{L}_j$ are the average losses for subgroups $i$ and $j$, respectively. We performed a grid search over $\lambda \in \{0.2, 0.25, 0.3, 0.35, 0.4\}$ using nested cross-validation. Each outer fold selected the optimal $\lambda$ based on the lowest EO gap observed across the corresponding inner folds. This strategy enabled us to reduce inter-group disparities.

### C. Bias Mitigation in LLMs

We leveraged in-context learning and examined two different prompting strategies: (1) guided prompting (GP), which explains linguistic patterns relevant to depression (e.g., self-focused pronouns, emotional tone); and (2) guided prompting with ethical framing (GP-EF), which builds on GP by adding explicit fairness-related instructions that emphasize

TABLE I
LLM PROMPT TEMPLATES USED IN FEW-SHOT IN-CONTEXT LEARNING

| Prompt Type | Description and Instructions |
|---|---|
| **Unguided Prompt (UP)** | You are an expert mental health assistant trained to detect signs of depression in conversations. Each prompt includes $1, \ldots, N$ *prior conversations*, followed by one test/target conversation. The model is instructed to predict the binary depression label (1 = depressed, 0 = not depressed) based solely on the conversation's language. The output format is strictly: `Test/Target Conversation corresponds to Label: [0 or 1]` |
| **Guided Prompt (GP)** | Same structure as UP, but includes a list of depression-related linguistic cues for the model to consider, such as: (1) Use of first-person singular pronouns (e.g., *I, me, my*), (2) Negative emotion words, (3) Verbosity and repetition, (4) Abstract/descriptive vs. analytical/concrete style, (5) Preference for past or continuous tense. The standard label format is enforced. |
| **Guided Prompt with Ethical Framing (GP-EF)** | Extends GP with explicit fairness guidance. The model is instructed to treat individuals equally regardless of socioeconomic status, gender, race, disability, or other protected characteristics. It is advised to avoid stereotypes and base predictions only on linguistic and behavioral cues. The standard label format is enforced. |

equitable treatment (Table I). We constructed $N$-shot prompts ($N \in 1, 3, 5$) stratified by gender and race to ensure balanced representation in the context window, and evaluated each configuration over 20 random trials using a consistent held-out test set.

### D. Evaluation Metrics

We computed balanced accuracy (BA), true positive rate (TPR/recall), precision, F1 score, and equalized odds (EO) to quantify both performance and fairness. Following [51], we defined EO as $1 - |\text{TPR}_{\text{non-sensitive}} - \text{TPR}_{\text{sensitive}}|$, where Male and Female participants are considered the non-sensitive and sensitive group, respectively, for assessing gender bias. For assessing race/ethnicity bias, White/Caucasian is the non-sensitive groups, while African American and Hispanic participants are the sensitive groups. To assess statistical significance of disparities in performance metrics (BA, TPR, Precision and F1 score), we conducted *t*-tests for binary attributes (i.e., Female/Male, African American/White, Hispanic/White) and analysis of variance (ANOVA) for multi-class attributes (i.e., race/ethnicity as a whole).

### V. RESULTS

### A. Depression Detection Bias in DNN-Embeddings and LLMs

Female participants exhibited a higher BA of 0.72 compared to males (0.66) with DNN embeddings before bias mitigation (Table II). This disparity is also evident in TPR, which depicts an absolute gap of 0.14 between the two groups. EO was 0.85, reflecting some disparity between the two groups. In terms of race/ethnicity, African American and Hispanic participants demonstrated a lower performance across all depression detection metrics compared to White/Caucasian participants (Table III). This resulted in relatively low EOs of 0.29 between Hispanic and White/Caucasian participants, and 0.75 between African American and White/Caucasian participants.

When using LLMs with UP, depression classification metrics and EOs by gender are higher than those of the DNN-based embeddings (Table IV). BA reaches 0.69 and 0.79 for male and female participants, respectively, and remains relatively stable with increased shots in $N$-shot learning (Fig. 1). TPR is 0.74 for males and 0.81 for females in the 1-shot setting, yielding an EO of 0.93, which is higher than that observed with DNN-based embeddings. While EOs continue to increase with 3- and 5-shot learning, this is accompanied by a reduction in TPR for female participants. Similar patterns

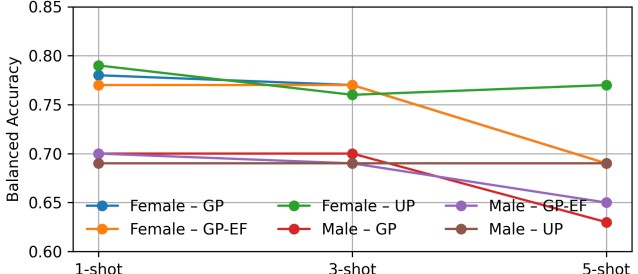

Fig. 1. Balanced accuracy by gender for 1, 3, and 5-shot learning across unguided prompting (UP), guided prompting (GP), and GP with ethical framing (GP-EF).

are observed when analyzing results by race/ethnicity, with LLMs using UP demonstrating improved depression detection performance compared to DNN embeddings before debiasing (Table V).

### B. Bias Mitigation in DNN-Embeddings

Applying fairness-aware objectives to DNN embeddings for bias mitigation led to increases in EO under both the worst-group and fairness-regularized loss functions for gender (Table II) and race/ethnicity (Table III). Both loss types significantly improved the model's ability to detect depression in the Hispanic subgroup, which had previously performed near chance levels. After debiasing, the classification performance and EO improved further for African American and Hispanic participants. When comparing the two loss functions, the worst-group loss achieves a better balance between classification performance and fairness across gender and racial/ethnic groups, while the fairness-regularized loss substantially increases EO, though at the cost of reduced accuracy.

### C. Bias Mitigation in LLMs

In the 1-shot setting, BA for male participants is 0.69 with UP and remains comparable under GP and GP-EF (Table IV). BA for female participants is higher, ranging from 0.77 to 0.79 depending on the prompt type. Differences in TPR between female and male participants are statistically significant under UP and GP but lose significance under GP-EF. This, along with the increased EO observed under GP-EF compared to UP and GP, suggests the effectiveness of GP-EF in mitigating gender bias. EO also increases from UP and GP to GP-EF in the 3-shot and 5-shot settings. However, *t*-test results indicate no statistically significant differences in TPR between genders across all prompting types. This may suggest that increasing

TABLE II
GENDER-WISE DEPRESSION DETECTION PERFORMANCE AND FAIRNESS METRICS FOR DNN-EMBEDDING

| Strategy | Group | TPR | Precision | F1 | Balanced Accuracy | Equalized Odds (EO) |
|---|---|---|---|---|---|---|
| Before debiasing | Male | $0.3864 \pm 0.0667$ | $0.6058 \pm 0.0488$ | $0.4650 \pm 0.0476$ | $0.6660 \pm 0.0267$ | 0.8598 |
| | Female | $0.5266 \pm 0.0869$ | $0.7336 \pm 0.1307$ | $0.6003 \pm 0.0545$ | $0.7252 \pm 0.0325$ | |
| Debiasing: Worst-group loss | Male | $0.7318 \pm 0.0100$ | $0.6892 \pm 0.0049$ | $0.7029 \pm 0.0023$ | $0.7049 \pm 0.0003$ | 0.9153 |
| | Female | $0.8165 \pm 0.0041$ | $0.6990 \pm 0.0006$ | $0.7448 \pm 0.0011$ | $0.7310 \pm 0.0008$ | |
| | $t(18) =$ | $-94.9711, p < 0.001$ | $-5.9263, p < 0.001$ | $-49.9689, p < 0.001$ | $-23.5115, p < 0.001$ | |
| Debiasing: Fairness-regularized loss | Male | $0.7696 \pm 0.0907$ | $0.5371 \pm 0.0633$ | $0.5922 \pm 0.0465$ | $0.5610 \pm 0.0284$ | 0.9551 |
| | Female | $0.8145 \pm 0.0785$ | $0.5744 \pm 0.0769$ | $0.6292 \pm 0.0412$ | $0.5912 \pm 0.0323$ | |
| | $t(18) =$ | $-1.0394, p = 0.3126$ | $-1.1214, p = 0.2776$ | $-1.7907, p = 0.0905$ | $-2.1658, p = 0.0443$ | |

TABLE III
RACE/ETHNICITY-WISE DEPRESSION DETECTION PERFORMANCE AND FAIRNESS METRICS FOR DNN-EMBEDDING

| Strategy | Group | TPR | Precision | F1 | Balanced Accuracy | Equalized Odds (EO) |
|---|---|---|---|---|---|---|
| Before debiasing | AA | $0.4600 \pm 0.1623$ | $0.7084 \pm 0.1989$ | $0.5143 \pm 0.1387$ | $0.7093 \pm 0.0722$ | HSP & WC = 0.2925 AA & WC = 0.7525 |
| | HSP | $0.0000 \pm 0.0000$ | $0.0000 \pm 0.0000$ | $0.0000 \pm 0.0000$ | $0.5000 \pm 0.0000$ | |
| | WC | $0.7075 \pm 0.0606$ | $0.5491 \pm 0.0501$ | $0.6168 \pm 0.0477$ | $0.7538 \pm 0.0190$ | |
| Debiasing: Worst-group loss | AA | $0.7417 \pm 0.1191$ | $0.5522 \pm 0.1210$ | $0.6239 \pm 0.0890$ | $0.7237 \pm 0.0550$ | HSP & WC = 0.7528 AA & WC = 0.8873 |
| | HSP | $0.8762 \pm 0.1524$ | $0.2414 \pm 0.1026$ | $0.3558 \pm 0.1173$ | $0.7809 \pm 0.0638$ | |
| | WC | $0.6290 \pm 0.1250$ | $0.6452 \pm 0.0696$ | $0.6256 \pm 0.0555$ | $0.6512 \pm 0.0446$ | |
| Debiasing: Fairness-regularized loss | AA | $0.9818 \pm 0.0364$ | $0.3761 \pm 0.0587$ | $0.5415 \pm 0.0631$ | $0.5651 \pm 0.0629$ | HSP & WC = 0.9775 AA & WC = 0.9957 |
| | HSP | $1.0000 \pm 0.0000$ | $0.1159 \pm 0.0550$ | $0.2034 \pm 0.0878$ | $0.5779 \pm 0.0659$ | |
| | WC | $0.9775 \pm 0.0287$ | $0.4928 \pm 0.0769$ | $0.6516 \pm 0.0694$ | $0.5697 \pm 0.0987$ | |

*AA = African American, HSP = Hispanic, WC = White/Caucasian.*

TABLE IV
GENDER-WISE DEPRESSION DETECTION PERFORMANCE AND FAIRNESS METRICS FOR LLM WITH VARIOUS PROMPTING STRATEGIES

| $N$-Shot | Prompt | Group | TPR | Precision | F1 | Balanced Accuracy | Equalized Odds (EO) |
|---|---|---|---|---|---|---|---|
| 1-Shot | Unguided Prompting (UP) | Male | $0.74 \pm 0.13$ | $0.38 \pm 0.03$ | $0.50 \pm 0.04$ | $0.69 \pm 0.04$ | 0.93 |
| | | Female | $0.81 \pm 0.10$ | $0.67 \pm 0.03$ | $0.73 \pm 0.03$ | $0.79 \pm 0.03$ | |
| | | $t(38) =$ | $1.87, p = 0.070$ | $32.79, p < 0.001$ | $19.11, p < 0.001$ | $8.44, p < .001$ | |
| | Guided Prompting (GP) | Male | $0.80 \pm 0.10$ | $0.38 \pm 0.03$ | $0.51 \pm 0.03$ | $0.70 \pm 0.03$ | 0.95 |
| | | Female | $0.85 \pm 0.09$ | $0.63 \pm 0.03$ | $0.72 \pm 0.02$ | $0.78 \pm 0.02$ | |
| | | $t(38) =$ | $1.75, p = 0.088$ | $25.81, p < 0.001$ | $23.26, p < 0.001$ | $9.06, p < .001$ | |
| | GP w. Ethical Framing (GP-EF) | Male | $0.78 \pm 0.10$ | $0.38 \pm 0.02$ | $0.51 \pm 0.03$ | $0.70 \pm 0.03$ | 0.97 |
| | | Female | $0.81 \pm 0.09$ | $0.63 \pm 0.04$ | $0.71 \pm 0.02$ | $0.77 \pm 0.02$ | |
| | | $t(38) =$ | $0.90, p = 0.377$ | $25.40, p < 0.001$ | $24.46, p < 0.001$ | $8.54, p < .001$ | |
| 3-Shot | Unguided Prompting (UP) | Male | $0.73 \pm 0.12$ | $0.37 \pm 0.03$ | $0.49 \pm 0.05$ | $0.69 \pm 0.04$ | 0.98 |
| | | Female | $0.71 \pm 0.12$ | $0.69 \pm 0.04$ | $0.69 \pm 0.06$ | $0.76 \pm 0.05$ | |
| | | $t(38) =$ | $0.59, p = 0.557$ | $25.82, p < 0.001$ | $12.05, p < 0.001$ | $4.91, p < .001$ | |
| | Guided Prompting (GP) | Male | $0.75 \pm 0.11$ | $0.37 \pm 0.02$ | $0.50 \pm 0.03$ | $0.70 \pm 0.03$ | 0.98 |
| | | Female | $0.77 \pm 0.13$ | $0.67 \pm 0.04$ | $0.71 \pm 0.06$ | $0.77 \pm 0.05$ | |
| | | $t(38) =$ | $0.36, p = 0.719$ | $30.68, p < 0.001$ | $13.49, p < 0.001$ | $5.76, p < .001$ | |
| | GP w. Ethical Framing (GP-EF) | Male | $0.76 \pm 0.10$ | $0.36 \pm 0.03$ | $0.49 \pm 0.04$ | $0.69 \pm 0.03$ | 1.00 |
| | | Female | $0.76 \pm 0.12$ | $0.66 \pm 0.04$ | $0.70 \pm 0.05$ | $0.77 \pm 0.04$ | |
| | | $t(38) =$ | $0.11, p = 0.912$ | $27.67, p < 0.001$ | $15.28, p < 0.001$ | $6.40, p < .001$ | |
| 5-Shot | Unguided Prompting (UP) | Male | $0.77 \pm 0.11$ | $0.34 \pm 0.04$ | $0.48 \pm 0.05$ | $0.69 \pm 0.05$ | 0.98 |
| | | Female | $0.79 \pm 0.14$ | $0.64 \pm 0.04$ | $0.70 \pm 0.06$ | $0.77 \pm 0.05$ | |
| | | $t(38) =$ | $0.59, p = 0.559$ | $22.53, p < 0.001$ | $12.49, p < 0.001$ | $5.71, p < .001$ | |
| | Guided Prompting (GP) | Male | $0.87 \pm 0.11$ | $0.29 \pm 0.05$ | $0.43 \pm 0.05$ | $0.63 \pm 0.07$ | 0.95 |
| | | Female | $0.92 \pm 0.09$ | $0.50 \pm 0.10$ | $0.64 \pm 0.07$ | $0.69 \pm 0.09$ | |
| | | $t(38) =$ | $1.37, p = 0.180$ | $8.39, p < 0.001$ | $10.56, p < 0.001$ | $2.05, p = 0.049$ | |
| | GP w. Ethical Framing (GP-EF) | Male | $0.88 \pm 0.12$ | $0.30 \pm 0.05$ | $0.45 \pm 0.06$ | $0.65 \pm 0.08$ | 0.99 |
| | | Female | $0.87 \pm 0.13$ | $0.52 \pm 0.10$ | $0.64 \pm 0.07$ | $0.69 \pm 0.08$ | |
| | | $t(38) =$ | $0.11, p = 0.916$ | $8.87, p < 0.001$ | $9.52, p < 0.001$ | $1.73, p = 0.094$ | |

TABLE V
RACE/ETHNICITY-WISE DEPRESSION DETECTION BALANCED ACCURACY (BA) AND EQUALIZED ODDS (EO) FOR LLM

| Race | 1-Shot | | | 3-Shot | | | 5-Shot | | |
|---|---|---|---|---|---|---|---|---|---|
| | UP | GP | GP-EF | UP | GP | GP-EF | UP | GP | GP-EF |
| AA | 0.84±0.06 / 0.74 | 0.84±0.04 / 0.76 | 0.84±0.05 / 0.74 | 0.81±0.08 / 0.73 | 0.82±0.05 / 0.75 | 0.81±0.04 / 0.78 | 0.81±0.06 / 0.80 | 0.74±0.13 / 0.85 | 0.74±0.11 / 0.86 |
| HSP | 0.69±0.04 / 0.87 | 0.68±0.04 / 0.90 | 0.68±0.03 / 0.87 | 0.69±0.05 / 0.88 | 0.68±0.05 / 0.92 | 0.67±0.04 / 0.96 | 0.63±0.06 / 0.99 | 0.59±0.07 / 0.96 | 0.62±0.09 / 1.00 |
| WC | 0.66±0.06 | 0.67±0.05 | 0.65±0.03 | 0.64±0.05 | 0.67±0.05 | 0.67±0.04 | 0.67±0.06 | 0.62±0.07 | 0.62±0.07 |
| ANOVA | $F(2, 57) = 15.23, p < 0.001$ | | | $F(2, 57) = 12.87, p < 0.001$ | | | $F(2, 57) = 11.02, p < 0.001$ | | |

*AA = African American, HSP = Hispanic, WC = White/Caucasian. UP = Unguided Prompting, GP = Guided Prompting, GP-EF = Guided Prompting with Ethical Framing. Results represent the mean and standard deviation of BA / EO values for AA and HSP relative to WC.*

the number of $N$-shots in LLMs could serve as an implicit debiasing strategy, although disparities persist across other evaluation metrics (Fig. 1). In terms of race/ethnicity, BA scores are the highest for the African American group across all configurations (Table V). ANOVA confirms that differences in BA are statistically significant across groups. The type of prompting or the number of examples in the $N$-shot learning do not impact the results when analyzed by race/ethnicity. While there are instances where EO increases with GP and GP-EF compared to UG (e.g., the 3-shot setting for both African American and Hispanic participants) these improvements are not consistent across the 1- and 5-shot cases.

### D. Cost-benefit analysis

The average running time to predict one sample was 0.0025s with DNN-based embeddings and 0.68s with GPT-4o-mini. This running time difference might be due to the difference between the two models in size (i.e., GPT-4o-mini has about 8 billion parameters, while the DNN-based model has 110 million parameters) and prompt construction overhead (i.e., the LLMs rely on auto-regressive token generation during inference that is computationally intensive for long text inputs, such as those in this study). Despite the fact that DNN-based embeddings exhibited a lower computational cost than LLMs, they exhibited lower depression detection performance and greater gender disparities in algorithmic fairness compared to LLMs. Bias mitigation methods further resulted in stronger debiasing for the LLMs compared to DNN-based embeddings.

## VI. Discussion

In addressing RQ1, models using DNN-based embeddings exhibit notable disparities in depression detection performance and fairness across demographic groups, despite being pre-trained on mental-health-specific data. We conducted a small-scale error analysis for the DNN-based embeddings. For example, a male participant (DAIC-WOZ ID: 340) was misclassified as depressed, potentially due to religious references (e.g., "*how evil the world is and your son has to live in it*") or expressions of fear (e.g., "*felt a little nervous, scared*") that were interpreted out of context. A Hispanic participant (DAIC-WOZ ID: 357) was misclassified with depression possibly due to informal language (e.g., "*normal kid stuff*") or vague phrasing (e.g., "*I don't even know*"). LLMs also exhibit disparities, though these are less pronounced between female and male participants, with EOs generally higher than in DNN-based models. Analysis by race and ethnicity is less conclusive. While overall classification improves significantly with LLMs, particularly for the Hispanic group, disparities between African American and White/Caucasian participants and between Hispanic and White/Caucasian participants persist, even as the number of examples increases in $N$-shot learning.

In answering RQ2, we observed a trade-off between optimizing bias mitigation and depression detection performance, as also noted in prior work [52]. The worst-group loss achieves a better balance compared to the fairness-regularized loss, potentially because it targets the subgroup with the poorest

performance. In contrast, the fairness-regularized loss enforces global fairness constraints that can disproportionately penalize performance in well-performing subgroups.

In addressing RQ3, GP did not provide notable improvements over UP for bias mitigation. However, GP-EF improved EO between female and male participants in the 1-shot setting. As the number of shots increased to 3 and 5, disparities between female and male participants decreased, with prompting strategy having minimal influence on the outcomes. In contrast, results by race/ethnicity indicate that neither prompting strategy nor increasing $N$ in $N$-shot learning effectively reduces disparities, as gaps in performance metrics remain largely consistent across groups.

This study has several limitations. First, although the overall sample size is relatively large, Hispanic participants were underrepresented among those with depression outcomes. This may be due to sampling bias during data collection, cultural or linguistic factors affecting the expression or reporting of depressive symptoms, or limitations in the sensitivity and validity of assessment tools across demographic groups. Second, we evaluated two representative models per category (i.e., MentalRoBERTa for DNN-based embeddings and GPT-4o-mini for LLMs). Future work should extend this analysis to a broader range of models to validate the generalizability of our findings, and explore personalized modeling approaches that capture individual-level differences in clinical profiles. Third, it remains unclear whether these results can generalize to other datasets with non-English speakers. A multilingual examination of algorithmic bias is beyond the scope of this paper due to space constraints, but we plan to include other datasets, such as the EATD-Corpus [53], in future work.

## VII. Conclusion

This study investigated algorithmic bias in language-based models for depression detection. While LLMs outperformed DNN-based embeddings overall, socio-demographic disparities, particularly across racial groups, persist. Bias mitigation in DNN embeddings using worst-group loss improved fairness without substantially reducing accuracy. In LLMs, GP-EF prompting improved fairness between male and female participants, but only in the 1-shot setting. These findings underscore the complexity of balancing fairness and accuracy in clinical NLP. They also highlight the need to tailor debiasing strategies to both model type and demographic context.

### Acknowledgment

The authors would like to thank Prof. Jonathan Gratch for providing meta-data from DAIC-WOZ, and the NSF ACCESS program for the computational resources.

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
