# OpenReview forum: "Assessing Algorithmic Bias in Language-Based Depression Detection: A Comparison of DNN and LLM Approaches"
_IEEE.org/EMBS/BHI/2025/Conference — BHI 2025_

### Official Review · Reviewer_NBbE · 2025-06-30
**Analysis of different methods of mitigating gender and ethnicity bias in depression prediction**

**Confidence:** 5
**Clarity Of Writing:** great
**Clinical Significance:** good
**Methodological Novelty:** fair
**Overall Rating:** 5
**Final Rating:** 6

**Experiments And Results:**

good

**Questions For The Authors:**

Include interpretability tools such as SHAP values, attention heatmaps or error analysis to understand why models make biased predictions.
Explore additional LLM architectures especially open-source options to generalize conclusions about fairness and performance. Also including the cost-benefit analysis of using LLM versus DNN for real-life application senarios.

**Strengths:**

This study addresses a crucial issue of bias in mental health NLP, highly relevant for ethical AI deployment. It thorouly compares DNN-based and LLM approaches on fairness and performance with very comprehensive metrics, including significance testing and multiple N-shots configurations.

**Summary Of The Paper:**

This study evaluates algorithmic bias in models used for automated depression detection from language data, focusing on disparities across gender and ethnicity. They compares traditional DNN-based embeddings to few-shot large language models. They explored fairness-aware loss functions to mitigate bias in DNN-based models, and asses different prompting strategies and number of shots for bias mitigation in LLM. They used the DAIC-WOZ dataset with 186 participants for gender analysis and 184 for race analysis. They found that LLM outperform DNN-based embeddings, especially among hispanic participants. They showed that the LLM has smaller gender disparity, but race disparity persists for both LLM and DNN. They also found that the worst-group loss function achieves better fairness-performance in DNN. In LLM, guided prompting with ethical framing improves gender fairness.

**Weaknesses:**

Limited generalizability that this study only focuses on the DAIC-WOZ containing only English-speaking clinical interviews. With previous studies showing the disparity in performance for other language, this study should also address fairness improvement on the other language settings.
This study only explored on LLM model that limits its broader insights. Moreover, with the LLM benefits of higher performance, it is also crucial to have cost-benefit analysis for using the computationally expensive LLM model vs the DNN-model.
The study did not include analysis of the model decision-making rationales or qualitative error analysis, which could provide insight into persistent disparities, making it lack interpretability.

---

### Official Review · Reviewer_sU3y · 2025-07-14
**Bias consideration**

**Confidence:** 3
**Clarity Of Writing:** great
**Clinical Significance:** great
**Methodological Novelty:** good
**Overall Rating:** 6

**Experiments And Results:**

great

**Questions For The Authors:**

The authors should clarify if there are any strategies used to handle imbalanced races.

**Strengths:**

This paper is well-written. The problem is clearly defined, and methods utilized are clearly presented.

**Summary Of The Paper:**

This work examines strategies to mitigate bias caused by race and gender bias in depression using data collected from clinical interviews. For DNN based models, fairness-aware loss functions were applied. For LLMs based models, in-context learning with different prompting schemes were explored.

**Weaknesses:**

The innovation of the presented work appears to be limited, as the fairness-weighted loss has been explored before.

---

### Official Review · Reviewer_Fmez · 2025-07-16
**Mitigating Algorithmic Bias in Language-Based Depression Detection Using DNN and LLM Approaches**

**Confidence:** 3
**Clarity Of Writing:** good
**Clinical Significance:** good
**Methodological Novelty:** excellent
**Overall Rating:** 8
**Final Rating:** 7

**Experiments And Results:**

good

**Questions For The Authors:**

1. Given the substantial inter-individual variability in clinical profiles, the authors are encouraged to elaborate on how the proposed pipeline addresses individual differences in lengthy interview transcripts and ensures that the DNN and LLM models support personalized predictions.

2. The fine-tuning procedures for the DNN and LLM models lack sufficient detail. In particular, it is unclear how the dataset was used for subject-specific fine-tuning. If the models were calibrated individually, how many transcript chunks per subject are necessary to achieve reliable prediction for the remaining content? Alternatively, if a leave-one-subject-out strategy was used, this should be explicitly stated. In either case, the validation approach should closely reflect realistic application scenarios to appropriately evaluate the model’s robustness, especially in light of substantial inter-individual variability.

**Strengths:**

This study evaluates the impact of algorithmic bias in language-based depression detection using clinical interview transcripts. The findings offer a valuable contribution beyond prior work that primarily focused on tabular and multimodal data. Additionally, the study is among the first to leverage LLM approaches for depression detection while addressing concerns of algorithmic fairness.

**Summary Of The Paper:**

This study addresses algorithmic bias in language-based depression detection. The authors investigate the effectiveness of two bias mitigation strategies applied within both deep neural network (DNN) and large language model (LLM) frameworks. The depression detection models, with and without bias mitigation, were implemented and evaluated using the DAIC-WOZ dataset.

**Weaknesses:**

While the manuscript is well-motivated and well-written, the fine-tuning phases and validation procedures for the DNN and LLM models are somewhat unclear and difficult to follow.

---

### Official Review · Reviewer_yDiS · 2025-07-18

**Confidence:** 4
**Clarity Of Writing:** good
**Clinical Significance:** good
**Methodological Novelty:** good
**Overall Rating:** 7

**Experiments And Results:**

good

**Questions For The Authors:**

See weakness

**Strengths:**

This paper studies an interesting problem and made comprehensive studies and comparisons among LLMs and DNNs. It shows the potential of DNN fairness in depression detection and maybe related areas, and also the limitations of existing debiasing methods of LLMs. The conclusions drawn from this paper are pretty interesting and useful.

**Summary Of The Paper:**

This paper studies the fairness problems of LLM and DNN methods in depression detection. In previous works, results showed that LLMs tend to have better overall performances than DNN methods. In this paper, the authors studies the fairness and biases of LLMs and DNNs. By certain debasing techniques, the authors show that DNNs can have less biases compared to LLMs even with common debias techniques.

**Weaknesses:**

A review of broader areas may help, such as in other detection problems.